# Glioblastoma Microenvironment and Cellular Interactions

**DOI:** 10.3390/cancers14041092

**Published:** 2022-02-21

**Authors:** Carmen-Bianca Crivii, Adina Bianca Boșca, Carmen Stanca Melincovici, Anne-Marie Constantin, Mariana Mărginean, Eleonora Dronca, Rada Suflețel, Diana Gonciar, Maria Bungărdean, Alina Șovrea

**Affiliations:** 1Anatomy and Embryology Discipline, Morphological Sciences Department, Iuliu Hațieganu University of Medicine and Pharmacy, 400000 Cluj-Napoca, Romania; bianca.crivii@umfcluj.ro; 2Histology Discipline, Morphological Sciences Department, Iuliu Hațieganu University of Medicine and Pharmacy, 400000 Cluj-Napoca, Romania; bianca.bosca@umfcluj.ro (A.B.B.); carmen.melincovici@umfcluj.ro (C.S.M.); mariana.marginean@umfcluj.ro (M.M.); rada.sufletel@umfcluj.ro (R.S.); simona.sovrea@umfcluj.ro (A.Ș.); 3Medical Genetics Discipline, Molecular Sciences Department, Iuliu Hațieganu University of Medicine and Pharmacy, 400000 Cluj-Napoca, Romania; eleonora.dronca@umfcluj.ro; 4Pathological Anatomy Discipline, Morphological Sciences Department, Iuliu Hațieganu University of Medicine and Pharmacy, 400000 Cluj-Napoca, Romania; diana.gonciar@umfcluj.ro (D.G.); maria.bungardean@elearn.umfcluj.ro (M.B.)

**Keywords:** glioblastoma, microenvironment, cellular interactions

## Abstract

**Simple Summary:**

This paper summarizes the crosstalk between tumor/non-tumor cells and other elements of the glioblastoma (GB) microenvironment. In tumor pathology, glial cells result in the highest number of cancers, and GB is considered the most lethal tumor of the central nervous system (CNS). The tumor microenvironment (TME) is a complex peritumoral hallo composed of tumor cells and several non-tumor cells (e.g., nervous cells, stem cells, fibroblasts, vascular and immune cells), which might be a key factor for the ineffective treatment since the microenvironment modulates the biologic status of the tumor with the increase in its evasion capacity. A deeper understanding of cell–cell interactions in the TME and with the tumor cells could be the basis for a more efficient therapy.

**Abstract:**

The central nervous system (CNS) represents a complex network of different cells, such as neurons, glial cells, and blood vessels. In tumor pathology, glial cells result in the highest number of cancers, and glioblastoma (GB) is considered the most lethal tumor in this region. The development of GB leads to the infiltration of healthy tissue through the interaction between all the elements of the brain network. This results in a GB microenvironment, a complex peritumoral hallo composed of tumor cells and several non-tumor cells (e.g., nervous cells, stem cells, fibroblasts, vascular and immune cells), which might be the principal factor for the ineffective treatment due to the fact that the microenvironment modulates the biologic status of the tumor with the increase in its evasion capacity. Crosstalk between glioma cells and the brain microenvironment finally inhibits the beneficial action of molecular pathways, favoring the development and invasion of the tumor and its increasing resistance to treatment. A deeper understanding of cell–cell interactions in the tumor microenvironment (TME) and with the tumor cells could be the basis for a more efficient therapy.

## 1. Introduction

The central nervous system (CNS) evolved in an integrated network composed of neurons (decisional cells) and neuroglial cells (homeostatic, immuno-, and activity modulators) [1,2]. In tumor pathology, glial cells result in the highest number of cancers [2].

Glioblastoma (GB), known as glioblastoma multiforme (GBM), is the most lethal tumor of the CNS, preserving its ranking as a grade 4 glioma since the beginning of the WHO classifications [3,4,5,6,7,8].

It seems that in GBM, the tumor microenvironment (TME) is the key factor impairing the efficacy of treatment. TME is a complex peritumoral hallo composed of tumor cells and several non-tumor cells (e.g., nervous cells, stem cells, fibroblasts, vascular and immune cells), occasionally prevalent [9].

The cell-to-cell interactions or the cancer cells themselves generate signals that activate the TME cells and determine, due to their high plasticity, alterations of their morphological, functional, and bioenergetic status (e.g., the pivotal role of mitochondria in modulating the GBM metabolism) [9,10]. A vicious circle is created. Upon activation, these cells are responsible for the secretion of a substantial number of inflammatory factors: cytokines, matrix metalloproteinases—MMP, growth and oxidative stress factors, that next will suppress or activate the signaling pathways, modifying their beneficial tissular action and promoting tumor development and invasion [11,12].

The metabolites secreted by the glioma cells alter the function of immune cells (glioma-associated microglia/macrophages—GAMs, natural killer—NK cells, cytotoxic T lymphocytes, and dendritic cells—DCs), promote the infiltration of immunosuppressive regulatory T cells and myeloid-derived suppressor cells (MDSCs) in TME and, thus, alter the recognition of the tumor. On the other hand, the overexpression of metabolic enzymes reduces the recruitment of NK cells and cytotoxic T lymphocytes while increasing the secretion of chemokines that attract inflammatory cells; the anti-tumor responses are hindered [10]. The metabolic rewiring thus involves all the TME’s various components activity [9]. As a consequence of tumoral growth in the peritumoral tissue, the metabolic changes trigger region-specific neuronal toxicity with neurodegeneration [9,13,14].

## 2. Interactions between Glioblastoma Cells and Tumor Microenvironment

### 2.1. Cell-to-Cell Communication

Tumors influence the microenvironment differently, transforming the physiological conditions into a suitable medium for their development [15]. In order to establish an immunosuppressive TME that allows the tumor progression, malignant cells from GBM communicate in a bidirectional manner with the normal brain cells from its surrounding environment. Almost all the cells from TME are involved in this process [16], with intercellular communication representing an essential feature for proliferation and metastasis. Similar mechanisms as in physiological communication are used, including either direct exchanges (via extracellular vesicles—EVs, gap junctions—GJs, ion channels, and transporters), or soluble factors (neurotransmitters, hormones, cytokines/chemokines, growth factors) [15]. Through all these factors, adjustments, and disruptions of the phenotype of vicinal and even distant cells are achieved [15].

Some of these mechanisms will be discussed below.

#### 2.1.1. Extracellular Vesicles—Carriers Passing through the Blood–Brain Barrier

Extracellular vesicles (EVs), membrane-bound structures without a nucleus [17,18], transport different types of cargo, such as bioactive and genomic material (i.e., DNA, mRNA, microRNA or miRNA), lipids, proteins, or nucleic acids, towards nearby and distant recipient cells, to influence their behavior [19,20]. Exosomes (30–120 nm), microvesicles (0.1–1 µm), apoptotic bodies (500–2000 nm), and large oncosomes (>1 µm) are all examples of EVs [19,21]. In the case of GBM and other malignancies, EVs can transport components between GBM cells and cells of TME [21]. In order to do so, GBM-derived EVs can pass through the blood-brain barrier (BBB) [22] because of Semaphorin3A present on their surface, which, by binding to the neuropilin1 receptors can disrupt the BBB (Figure 1) [23]. By transporting and releasing different types of cargo, EVs play an active part in the malignant behavior of GBM, such as invasiveness and tumor progression, angiogenesis, and drug resistance [24,25,26]. Proliferation and migration are favored by transporting angiogenic proteins (VEGF-A) and RNA in GBM TME [27,28]. Furthermore, when EVs cargo is VEGF-A, vascularization in the TME of GBM is stimulated and maintained [29]. Sun et al. studied the angiogenesis phenomena in gliomas and revealed that EVs (produced from GSCs), which transport miRNA encoded by the *MIR21* gene (miR-21), increase endothelial cells migration and tubular structure formation; he reinforced this statement by showing that the suppression of VEGF leads to the reduction in endothelial cells tube formation, implying the VEGF involvement in the mediation of miR-21-associated neo-angiogenesis [30]. On the other hand, the EVs implicated in miR-21 transport between GBM cells and TME has a different effect, reducing angiogenesis, tumorigenicity, and invasion, and thus suppressing malignant tissue [31]. Hypoxia seems to be an influencing factor in cellular communication between malignant cells and TME. GBM EVs cultivated under hypoxic circumstances were shown by Kucharzewska et al. to modify the phenotype of endothelial cells, in order to induce angiogenesis [32]; hypoxic GBM cells can induce paracrine activation of endothelial cells via EVs with pro-angiogenic protease-activated receptor-mediated heparin-binding EGF signaling [33]. EVs from GBM also impact the PD1/PD-L1 pathway by potentially stopping T cells from becoming activated and proliferating [34]. By preventing T-cell infiltration, the growth and invasiveness of GBM are supported [35]. As researched by Ding et al., in glioma cell lines, hypoxia upregulates PD-L1 expression via HIF-1α [36].

Altogether, EVs’ bidirectional transport between GBM and its TME is essential in tumor angiogenesis, invasiveness, and growth but also in the disruption of the BBB.

#### 2.1.2. Gap Junctions and Their Role in Cell Communication

Gap junctions (GJs) represent another way by which astrocytes can communicate with GBM cells: in glioma-associated astrocytes, the GJ protein connexin-43 (CX-43) increases chemotherapy resistance, GBM’ cells proliferation, and migration [37,38].

GJs are involved in cellular communication and contribute to cell survival [39]. GJs comprise integral membrane proteins called connexins (CX), which allow for the bidirectional movement between cells of various elements (e.g., ions, miRNAs, second messengers), and other small molecules [40,41]. By doing so, GJs are implicated in cell development and differentiation, and overall tissue homeostasis [40,41].

Adhesion complex disruption is also involved in cancer pathogenesis [42]. The relationship between GBM cells and GJs plays a vital role in the TME of this brain malignancy. Osswald et al. studied how brain tumor cells connect and communicate and revealed the role of CX-43-based GJ, which can interconnect GBM cells to a multicellular network, that can communicate over long distances (thus, GBM cells can infiltrate healthy cerebral tissue) [43]. Furthermore, this mechanism can contribute to the therapeutic resistance of these tumors [44]. However, when comparing the expression of CX-43 of GBMs to healthy brain tissue, Soroceanu et al. and Pu et al. observed a decreased expression of CX-43 in higher grade gliomas [45,46]. Taking into consideration the fact that malignant gliomas produce increased epidermal growth factor (EGF) [47], the decreased expression of CX-43 can be explained by the phosphorylation of CX-43 by EGF, or by lysophosphatidic acid via a mitogen-activated protein kinase, leading to the disruption of GJs communication [48]. Potthoff et al. showcased the long-distance communication between GBM cells and the multicellular network, by staining for CX-43, and observing positivity along the thin cell protrusions, which connect all cells [49]. Astrocytes and glioma cells express CX-43 through their GJs (Figure 2) [45,50]. CX-43-mediated GJ coupling between glioma cells and astrocytes partly explains the involvement of astrocytes in GBM TME by changing their phenotype, and thus creating a more permissive environment for GBM invasion [51].

By understanding the role of GJs in TME, new therapeutic agents that can inhibit cell communication via GJs can be developed.

#### 2.1.3. Ion Channels and Transporters—Influencing GBM Cell Communication, Polarization, Shape, and Size

Ion channels and transporters such as hydrogen (H^+)^, potassium (K^+)^, and calcium (Ca^2+^) are another way of communication between cells from GBM and reactive astrocytes, employing ion concentration changes, cell volume variations, a loss of glioma cells’ epithelial polarization, or an increase in their migratory capacity (Figure 3) [38].

Channels involved in the transfer of sodium (Na^+)^, K^+^, and Ca^2+^ ions are one of the pathways frequently damaged in GBM, because ion channels are expressed in glial cells in various ways [52,53]. Ion channels can favor GBM invasiveness, by altering ion and water transport via the cell membrane, thus, resting membrane potential, and facilitating cell shape and volume’ changes [54].

One of the channels involved in GBM is the Ca^2+^- activated K^+^ channels, which respond to alterations in Ca^2+^ concentration; thus, the intracellular increase in Ca^2+^ determines a more negative potential of the channel.

While comparing malignant glioma tissue to nonmalignant tissue, overexpression of big conductance (B) K^+^ channels was observed and linearly correlated with glioma grade [55]. Those channels respond to intracellular and membrane voltage potential, by alternatively splicing their α-subunits. A novel splice isoform of hSlo, the gene that encodes the subunits, has greater sensitivity to intracellular Ca^2+^. Glioma is the only place where this BK channel isoform has been found. Furthermore, because the classical BK channel has yet to be discovered in gliomas, glioma is most likely expressing only this novel isoform [54]. GBM cells require Ca^2+^ as a second messenger to facilitate cell movement. It has been discovered that oscillatory fluctuations in intracellular Ca^2+^ correspond with cell invasion and migration [54]. Ca^2+^ permeable alpha-amino-3-hydroxy-5-methyl-4-isoxazolepropionate (AMPA) glutamate receptors are expressed in GBM cells [54]. Ishiuchi et al. studied Ca^2+^ permeable AMPA receptors in GBM cells, and stated, that due to the lack of the GluR2 subunit in GBM cells, these glutamate receptors had become Ca^2+^ permeable, emphasizing that GluR1 and GluR4 subunits are ubiquitously expressed in human GBM cells. AMPA receptors with GluR2 subunits have low Ca^2+^ permeability, whereas those without GluR2 subunits, have high Ca^2+^ permeability [56]. Ishiuchi et al. transferred GluR2 cDNA through an adenovirus, resulting in: reduced intracellular [Ca^2+^], hindered cell motility, and induced apoptosis [56]. Changes in the resting membrane potential of GBM cells can also be due to the ether-à go-go 1 (Eag1) and ether-à-go-go related 1 (Erg1), K^+^ channels, and members of the voltage-gated K^+^ channel family. Studies have shown an increase in the expression of these channels in GBM tissue, hence their role in glioma genesis [57]. Furthermore, Erg1 activity has also been linked to the activation of VEGF secretion, implying that it plays a role in angiogenesis [57] and neo angiogenesis [58].

Regarding NA^+^ channel mutations, researchers have different opinions about the association with *IDH1* mutations [59]; however, studies show a shorter survival period for patients with NA^+^ channels mutations [60]. In a study by Joshi et al., NA^+^ channel inhibitors (e.g., digoxin and ouabain), were given to two GBM cell lines, and the proliferation of GBM cells was investigated [60]. Both medicines had antiproliferative and cytotoxic effects on the cell lines and showcased an apoptotic phenotype under light microscope examination [60]. This study emphasizes the critical role of NA^+^ channels in cell-to-cell communications of GBM TME, and promotes new therapeutic methods for these aggressive tumors.

### 2.2. Dynamic Shape-Shifting Cellular Process Influencing GBM Characteristics

#### Epithelial-to-Mesenchymal Transition

Epithelial-to-mesenchymal transition (EMT) is a reversible cellular process, which implies the transitions of epithelial cells into mesenchymal cells’ states, in this manner currently playing an important part in embryogenesis and wound healing [61,62]. EMT entails interaction between cancer cells and immune cells, and between cells and the extracellular matrix, which can lead to changes in: cell polarity, loss of cell adhesion, increased migratory ability, shape-change, and changes in chemo response [61,63,64].

EMT is defined by the dynamic transition phases between epithelial and mesenchymal phenotypes, passing through an intermediate phase in which cells have both epithelial and mesenchymal features (the intermediate phase has great importance in fibrosis and tumor progression) [9,11]. The transition of epithelial cells into mesenchymal cells is defined by a gradual acquisition of motile and invasive behavior, accompanied by a change in gene expression that leads to the loss of epithelial properties and the acquisition of mesenchymal ones (Figure 4) [64,65,66,67,68,69,70]. The opposite of EMT is the mesenchymal-to-epithelial transition (MET), and it implies the loss of migratory flexibility and cells’ regain of apicobasal’ polarization and genes associated with epithelial’ cell phenotype [64,71].

There are three types of EMT: type 1- specific for embryogenesis, generating morphological and functional distinct cell types [67]; type 2- associated with regeneration processes, when fibroblasts appear in the injured tissue, also possibly occurring before the onset of tumorigenesis [68]); type 3- the only found in cancer cells that suffer: phenotypic conversion, increasing migration, invasion, and metastasis. There are studies that suggest that EMT is under control of transforming growth factor (TGF)-β through Smad or p38 mitogen-activated protein kinase/Ras homolog family member A pathways [69,70].

Although EMT has been first described in epithelial tumors, recent studies evaluate its connection to GBM’ progression, invasiveness, and chemotherapy resistance [72,73], emphasizing its role in glioma-genesis and remodeling of glioma TME [74]. When neoplastic cells undergo EMT, new-formed mesenchymal cells interact with the non-neoplastic cells (e.g., immune cells), altering their activities and representation in the TME [75].

In GBM, the hypoxic microenvironment can induce the EMT, as a secondary phenomenon, after the recruitment of the residential or circulating myeloid cells, microglia, and macrophages as well [70]. The release of growth factors (TGF-β, EGF, platelet-derived growth factor—PDGF, and fibroblast growth factor-2—FGF-2), cytokines and chemokines by these cells, trigger the alterations of the transcription factors, which initiate the EMT [70,76,77]. The modified expression of the specific transcription factors (such as: ZEB1, ZEB2, SNAI1, SNAI2, TCF, or miRNA) [61,78] will determine the location’ loss of the epithelial marker known as E-cadherin, and the levels’ augmentation of mesenchymal markers (such as: vimentin, N-cadherin, fibronectin, alfa-smooth muscle active), which will lead to the loss of cell adhesion [79,80].

The loss of E-cadherin (member of the superfamily of cadherins- adhesion molecules, essential in cell adhesion and homeostasis [81], that act, depending on the tumor-associated setting, either as tumor suppressors or promoters [61,82]) is the main event of EMT. E-cadherin can be directly repressed by these factors, that bind to the E-cadherin promoter, inhibiting its transcription. At the same time, there are factors (such as: Twist, Goosecoid, TCF4, FOXC2, and SIX1t), which indirectly repress E-cadherin [83].

Considering that astrocytes and malignant GBM cells are not typical epithelial cells, the classical EMT model can be altered. In the literature, there is inconsistency regarding the expression of E-cadherin in GBM, with some papers claiming that GBMs do not express E-cadherin, and others claiming that E- and N-cadherin flipping occurs [61,84,85]. Cadherin expression in cells can be heterogeneous, with cells expressing numerous cadherin subtypes, resulting in cadherin-mediated heterotypic adhesion [86]. In gliomas, the expression of E-cadherin is most commonly absent or scant [87,88], and if it is present (GBM subtypes with epithelial and pseudo-epithelial differentiation), it is usually correlated with a worse prognosis [37,89]. E-cadherin inconsistent expression can be caused by tumoral heterogenicity, translated in variations of gene expression, as suggested by The Cancer Genome Atlas (TCGA) data.

Of all cadherins present in the nervous tissue (N-cadherin, cadherin-11, cadherin-6, cadherin-8, or M-cadherin), N-cadherin is the most expressed. Regarding its expression in GBM tissue, it was found that 60 to 80% of all GBM express N-cadherin [90]. The down-regulation of N-cadherin in GBM has been linked to the aberrant cell polarization and motility, as well as to a considerable increase in tumor cell’ migration and invasiveness [90,91]. Siebzehnrubl et al. found that GBM cancer stem cells’ invasiveness is promoted by redistribution of N-cadherin’s anchoring to the cytoskeleton by ROBO1 [73]. On the other hand, upregulation of N-cadherin has been shown to rise alongside the glioma Ki-67 index, implying that cell adhesion’ signaling is involved in tumor cell’ proliferation and dedifferentiation [37,88,92].

Based on the genomic abnormalities extensively described by TCGA, Verhaak et al. [93] molecularly classify GBM into four gene expression subtypes as follows: classical, mesenchymal, proneural and neural. These subtypes show distinct differentiation characteristics that may translate into targeted therapies in the future [93]. Even single cell studies show GBM heterogenicity: based on Verhaak’s classification, Patel et al., using single cell RNA-seq, researched individual cells from five primary GBs and discovered that individual cells from different GBs subtypes are mixed together in a heterogeneous combination in each of the studied tumors [94]; most often a single cell scored well in the duo classical and proneural forms or mesenchymal and neural subtypes.

## 3. Tumoral and Reactive Astrocytes

When brain malignancies develop, the peritumoral tissue becomes enriched with astrocytes [95].

Current knowledge on the cellular origin of malignant gliomas points to three lineages: neural stem cells, glial progenitors, and astrocytes; each of these cell types, under particular conditions, transform to promote gliomagenesis. Clinical studies have demonstrated that gliomas with a more pronounced astrocytic phenotype are more aggressive and have a worse prognosis [96].

As part of the GBM microenvironment, the stromal cells, among which astrocytes are numerous, seem to play crucial roles in tumor maintenance, its progression, and its resistance to treatment [97]. Even though the morphology of the stromal astrocytes has been described, their biological activity is not entirely understood.

Astrocytes are implicated in multiple mechanisms associated with the development and progression of GBM: they can be cells of origin for these tumors [96], but they can also be non-neoplastic, stromal cells present in the TME.

Since astrocytes can be cells-of-origin for GBs, their heterogeneity influences the tumor transformation. Thus, the oncogenic TRP mutations in different subpopulations of astrocytes can lead to the formation of various types of gliomas [98]. Irvin et al. demonstrated that the mutations in astrocytes that expressed GFAP induced the formation of anaplastic astrocytoma, whereas the mutations in astrocytes that expressed glutamate/L-aspartate transporter (GLAST) gave rise to low-grade astrocytoma [99].

Astrocytes’ morphological and functional heterogeneity plays an important role in their tumorigenic potential, and the identification of astrocyte’ subpopulations, based on marker combinations, could differentiate normal astrocytes from their malignant counterparts [100].

Katz et al. compared the gene expression of tumor-associated astrocytes in low and high-grade GBM and described a subpopulation of astrocytes associated with stem-like glioma cells in the perivascular niches. These astrocytes express osteopontin, a ligand that enables the interaction with CD44^+^ glioma cells and has been correlated with poor prognosis [97]. Moreover, the perivascular tumor-associated astrocytes that express phosphorylated PDGF Receptor β promote the metastatic growth of glioma [101]. Astrocytes in the peritumoral areas express the Glial Cell Line-Derived Neurotrophic Factor (GDNF) to facilitate the invasive tumor growth [102].

Histomorphologically, in routine hematoxylin and eosin (H&E) stain, the neoplastic astrocytes were described as undifferentiated “naked” cells, containing a low amount of cytoplasm, practically undetectable, and characterized by large, elongated, pleomorphic and heterochromatic nuclei. Characteristic for the high-grade gliomas, the neoplastic cells exhibited a high nuclear/cytoplasmic ratio. Contrarily, the reactive astrocytes were larger and star-shaped, with abundant cytoplasm and processes with symmetrical distribution [103].

The tumor-associated astrocytes are an essential component of the gliomas, capable of interacting with GB cells, and influencing the tumor behavior by performing immunological functions [97], and becoming activated [104] (Figure 5).

In addition to their structural and functional roles in the BBB, and the transmission of nerve impulses, astrocytes are implicated in the response of the CNS to injury. Under pathological conditions, reactive astrocytes undergo morphological and functional changes through a process called astrogliosis, proliferate and migrate to the injury site. Moreover, reactive astrocytes can influence nervous tissue regeneration, either by stimulating or inhibiting neurogenesis, depending on the type of injury [105,106].

Traditionally, astrocytes have been identified by the Glial Fibrillary Acidic Protein (GFAP) expression, which is significantly higher in reactive astrocytes. However, the expression of GFAP has also been demonstrated in adult neural stem cells (aNSCs) in the adult brain. These stem cells are located in the subventricular zone associated with the lateral ventricles, and in the sub-granular zone in the hippocampus—the dentate gyrus—[107], and can give rise to progenitor cells that proliferate and differentiate to form neurons and glial cells [108]. In the context of injuries, reactive astrocytes exhibit plasticity, demonstrated by structural and functional characteristics common to aNSCs, including the expression of specific markers, and the ability to form neurospheres [106,109,110]. Since reactive astrocytes have stem cell properties, non-neoplastic astrocytes within the glioma could be activated to promote tumor growth and progression [97].

The astrocytes in GBM become activated under the influence of both the innate immune system and the tumor [95].

One of the mechanisms for the activation of astrocytes implicates the crosstalk with the microglial cells. Since astrocytes cannot respond to inflammatory factors, such as bacteria-derived toxins (e.g., lipopolysaccharides), the microglial signaling is essential for astrocyte activation [95]. The behavior of astrocytes can also be influenced by tumor cells, in order to promote glioma progression. Cytokines released by tumor cells, such as IL-10 and IFN-β, can manipulate the immunological functions of astrocytes by inducing the anti-inflammatory astrocytic phenotype [98].

The complex microglia–astrocytes–tumor cells crosstalk is mediated by multiple factors, including: (i) alterations of EMC and TME components due to tenascin-C, osteopontin, lactadherin, Fibulin-3, released by tumor cells, promote tumor growth and invasion [111]; (ii) reorganization of microtubules and F-actin in cytoskeleton increase the migratory capacity of microglial cells [111]; (iii) cytokines such as VEGF, IL-10, IL-6 and TGF-β induce an anti-inflammatory environment in the glioma, favorable for tumor progression [112]; (iv) chemokines and receptors expressed by microglia and astrocytes, such as CX3CL1 and its receptor CX3CR1, intermediate the communication between microglia, astrocytes and tumor cells [113]; (v) neurotrophic and morphogenic factors that promote the development of neurons and also control the activity of microglia and astrocytes, including GDNF, TGF-β, VEGF, EGF and the colony-stimulating factor-1 (CSF-1) [112,114,115]; (vi) metabolic factors associated with hypoxia and the alterations of glucose metabolism lead to microglia polarization and astrocyte activation [116,117]; (vii) dysregulation of miRNAs plays a role in the interactions between the cells and TME that lead to gliomagenesis [118].

The GBM-associated reactive astrocytes become hypertrophied, and the morphological changes lead to the differentiation of multiple subtypes. The immunolabeling for GFAP indicated a gradual change in astrocyte morphology and density during the GBM progression [119]. In the early stages of malignant transformation, reactive astrocytes are often in mitosis; morphologically, these astrocytes exhibit a round cell body and express GAFP and Nestin. In more advanced stages of GBM, reactive astrocytes have an enlarged cell body and extend long and thick processes and are characterized by the overexpression of GFAP [120]. In the case of the non-invasive GBM, the growing tumor is surrounded by an area enriched in reactive astrocytes that are organized in a sheet-like arrangement to form an astrogliosis capsule. In the zone of tumor invasion, reactive astrocytes form a dense network due to the maladaptive plasticity of the CNS and the subsequent changes in the composition of the microenvironment [121,122].

The interactions between the end-feet of the astrocytes and pericytes and endothelial cells are essential for the proper function of the BBB, including the delivery of nutrients and drugs to the CNS. Tumor cells cause the disconnection between astrocytes and pericytes and alter the normal function of the BBB, leading to drug resistance. Kim et al. studied the signaling pathway between astrocytes and the endothelial cell barrier in mice in an orthotopic model of human GBM, and demonstrated the protection of cancer tissue from chemotherapeutic agents given by these cells [123]. Moreover, the resulting perivascular spaces provide pathways for tumor growth and invasion [124].

During GBM progression, the disruption of the BBB enables the entrance of immune cells that activate the astrocytes and promote neuroinflammation [38]. Evidence has demonstrated a positive loop between astrocytes and microglia at the tumor site: the microglial cells release IL-6 that activates the peri-tumoral reactive astrocytes to secrete Monocyte Chemotactic Protein-3 (MCP-3) via JAK-STAT proteins, which further attracts more microglial cells [125].

Recent studies described specific metabolic changes in reactive astrocytes associated with GBM [9]. Under normal conditions, astrocytes have a glucose-based metabolism: they uptake the blood glucose via transporters (GLUTs); then, in the astrocytes, glucose can be either stored as glycogen or metabolized by glycolysis to pyruvate, which is further converted to lactate by the lactate dehydrogenase [117]. Reactive astrocytes are capable of modulating their metabolism in response to environmental changes [126]. The insufficient glucose levels, caused by the increased metabolic demands of the GB, trigger glycogenolysis, and the resulting metabolites are transferred to neurons to maintain the energetic neuronal metabolism [127]. The decrease in glucose levels can alter the mitochondrial function, leading to excessive production of reactive oxygen species (ROS) [128].

Moreover, a metabolic shift from glucose to glutamate or ketones can occur in astrocytes to produce energy by gluconeogenesis or ketosis [129,130,131]. These metabolic changes suppress the tumor progression by anti-angiogenic, anti-inflammatory, and pro-apoptotic effects [132]. Hypoxia, caused by the intense astrocytes’ metabolism and the ischemia, due to the compression of blood vessels at the periphery of the GB lead to intense glycolysis, with high levels of amino acids and nucleotides, such as ATP, which are transferred to tumor cells by CX-43 [133].

In contact with GBM cells, astrocytes become reactive and secrete a wide range of cytokines and chemokines that modify the TME and further promote astrocytes’ activation and the interactions between astrocytes and glioma. The upregulation of GFAP and CX-43 in astrocytes, and the expression of MMP-2 by the tumor cells, promote tumor infiltration [70,134]. Additionally, reactive astrocytes secrete growth factors, such as stromal cell-derived factor-1, supporting tumor cell proliferation and malignant transformation [135].

Several studies demonstrated the pathways implicated in the interactions between astrocytes and glioma cells [14]. Invasive GBM expresses high levels of NF-κB ligand (RANKL) and activates astrocytes at the periphery of the tumor; reactive astrocytes are stimulated, through NF-κB signaling, to secrete TGF-β that promotes tumor progression [136]. Clement et al. demonstrated that hedgehog (HH)-Gli signaling controls the expression of stemness genes in GBM stem cells and contributes to the activation of astrocytes in the perivascular niches in the tumor proximity; thus, it favors tumor growth and invasion [137]. Mutation of tumor suppressor gene *TP53*, which occurs in most GBM, is responsible for inhibiting the apoptosis in cancer cells, leading to increased malignancy. Moreover, the GBM cells block the expression of *TP53* in astrocytes and modulate the secretion of proteins that change the composition of TME and trigger the EMT, resulting in a more migratory and invasive phenotype [138,139]. Tumor-associated astrocytes secrete IL-6 that activates JAK/STAT signaling, which is linked to the more aggressive progression of GBM [140]. Astrocyte elevated gene-1 activates the PI3K/Akt pathway, which induces GBM cell proliferation and is associated with the grade of malignancy [141].

The high growth factors, cytokines, and other substances secreted by reactive astrocytes and exposed to the GBM microenvironment initiate various mechanisms to support tumor growth [14]. TNF-α, TGF-β, insulin growth factor-1 (IGF-1), and VEGF, released by astrocytes in the tumor proximity, promote GBM cells proliferation and invasion [142]. L-glutamine (Gln) is essential for the physiological balance between carbon and nitrogen in the nervous tissue. Gln-addiction is considered a characteristic of GBM, one of the most aggressive brain tumors [143]. The amount of Gln produced by tumor cells is insufficient for the metabolic needs of the developing tumor. Tardito et al. demonstrated that astrocytes uptake glutamate (Glu), synthesize Gln by using glutamine synthetase to convert Glu to Gln, and secrete Gln to supplement the TME, thus enabling the growth of glutamine-restricted glioblastoma cells [144]. Peritumoral astrocytes express high levels of CX-43, a significant protein in the GJs involved in the heterocellular communications between tumor cells and glioma-associated astrocytes. CX-43 is responsible for forming an invasive niche and for the spreading of glioma cells into the brain parenchyma [145].

The factors secreted by astrocytes also alter the TME to facilitate GBM invasion. IL-6, secreted by glioma-associated astrocytes, promotes the activation of proteolytic enzymes, the MMPs (including gelatinases MMP-2, MMP-9, and MMP-14) involved in tumor invasion and metastasis, by the remodeling of the extracellular matrix and the degradation of type IV collagen of the endothelial basement membrane [146]. IL-6 also upregulates the expression of fascin-1 (an actin filament bundling protein) that regulates cytoskeletal structures, resulting in the formation of protrusions related to cell motility for the migration and increased invasion potential of tumor cells [147].

Reactive astrocytes could be involved in the immune protection of tumor cells by secreting various factors. High levels of tenascin-C in the tumor extracellular matrix have an inhibitory effect on T cells migration [148]. Reactive astrocytes and tumor-associated macrophages, which lack phagocytotic activity, secrete IL-10, which has anti-inflammatory properties (by inhibiting the expression of MCH—major histocompatibility complex class II and IFN-γ). IL-10 also reduces the anti-tumor activity of T cells and NK cells, thus promoting tumor growth [149]. Additionally, upregulation of STAT-3 in reactive astrocytes promotes angiogenesis, immunosuppression, and tumor invasion [150]. STAT-3 activates myeloid-derived suppressor cells that expand during cancer progression and suppress the T cells response [151]. STAT-3 also inhibits the differentiation of immature myeloid cells into dendritic cells, macrophages, and microglia [152]. Therefore, reactive astrocytes can activate several immunomodulatory mechanisms, supporting the immune evasion of GBM.

Reactive astrocytes associated with GBM could also be responsible for the resistance to chemo-radiotherapy. Numerous studies demonstrated that the interaction of GBM cells with the microenvironment, particularly with the tumor-associated astrocytes, could play essential roles in this resistance [153]. The astrocytes associated with GBM protect the tumor cells from the apoptotic effect of chemotherapeutics by two mechanisms: (i) the GJs, connecting astrocytes to tumor cells that allow Ca^2+^ ions sequestration and miRNA transfer, and (ii) the endothelin receptor signaling pathway, due to the connections between astrocytes and endothelial cells [123,154,155,156,157]. The communications between astrocytes and tumor cells by GJs could decrease the sensitivity of tumor cells to chemotherapy [158]. Astrocytes could also repair DNA double-strand breaks caused by radiotherapy due to several gene expression profiles, including STAT-3 and Akt [159,160].

The broad functional diversity of astrocytes in the human brain suggests that distinct subpopulations of astrocytes could perform diverse roles. Moreover, their implication in various neurological pathologies, including brain malignancies, may be explained by selective changes in specific disease-associated phenotypes [161]. John Lin et al. identified discrete subpopulations of astrocytes in glioma and investigated their dynamics and roles in tumor progression [161]. The emergence of specific populations of pathological astrocytes during glioma progression was associated with tumor invasion, increased hyperexcitability, and seizure onset. The molecular profile of a subset of astrocytes indicated the expression of epilepsy-associated genes, suggesting that tumor-driven epilepsy could have both a neuronal and glial basis [161]. Moreover, specific subpopulations in the glioma alter the neuronal microenvironment, leading to increased synaptic activity, which, by feedback, promotes cell proliferation and tumor growth [162].

As essential contributors in the GBM microenvironment, reactive astrocytes perform multiple roles that promote the tumor cells’ survival, proliferation, invasion, immune evasion, and resistance to therapy. Further understanding of the pathogenic mechanisms underlying the interplay between GBM cells and reactive astrocytes could improve the therapeutic strategies’ efficacy in brain malignancies.

## 4. Glioma Stem Cells

Initially, the notion of cancer stem cells outlined the idea that the tumor derives from one or more cells who suffered mutations, subsequently dividing rapidly and forming the tumor as a clone of the initiator cell. However, there were distinguished subsets of heterogeneous tumor cells [163,164,165], with different cells proportions, histological appearances, procreation rates, expressions of surface markers, metastatic potentials, and responses to chemotherapy. This tumor heterogeneity was also tried to be explained through co-existence of several tumor stem cells that differentiate and create the types of cells that make up the tumor [166,167,168].

Several controversial and not yet clarified aspects confuse the glioma stem cells (GSCs) definition and identification: their origin, their genotype and phenotype, their continuous dynamic (reshaping through crosstalk with TME or by trans-differentiation). GSCs number inside the tumor mass is low and the multitude of GSCs’ regulatory mechanisms (genetic, epigenetic, metabolic, immune and environmental) increase the difficulty to define the GSCs [169].

The first controversies involved the GSC definition itself. The GSCs are cells that: (1) are able to initiate the tumor after a serial transplantation; (2) have the capacity to self-renew; (3) are able to reiterate the tumor cell heterogeneity [170]. Parallel with the stemness decrease, this functional definition of GSCs allows both the rigidity and plasticity hierarchical models of cell division [170].

Secondly, the source of tumor stem cells is controversial: either they originate from normal neural stem cells that undergo mutations, or they originate from the differentiation of transformed cells [171,172]. According to Feinberg’s theory, the origin of cancer stem cells may be the adult stem cells, with a lifespan that allows them to acquire epigenetic alterations [173]. The cell division of these tumor-initiating cells is asymmetric: each stem cell forms two daughter cells: one, undifferentiated, maintains the stem cell pool, and the other, a progenitor cell, differentiates and generates a wide range of tumoral cells [168]. Defects in asymmetric cell division of neuronal stem cells contribute to neoplastic transformation [174] and GSCs generation. The GSCs can then differentiate into various phenotypical cells—neurons, astrocytes, or oligodendrocytes and induce tumors after transplantation [169,170]. It is still possible that the cellular heterogeneity of GBM has an origin based on mono- or polyclonal GSCs [169].

Human GSCs were first identified by Singh et al. in an in vivo xenografted assay, and GSCs were able to initiate tumor growth [175]. Initially, the GSCs were reported to originate from neural stem cells (NCS) of the subventricular zone, in contact with a vascular niche; here, the factors secreted by the endothelial cells could induce the persistence of a stem-cell-like state [176]. A recent study using single-cell sequencing technique followed by laser microdissection analysis showed that GSCs could also originate from astrocyte-like neural stem cells of the subventricular zone that display low-level driver mutations and generate differentiated gliomas [170,177]. Another source for GSCs can be represented by the GBM cells, which can differentiate into GSCs in different stress conditions of TME, through deregulating signaling pathways such as SHH and WNT [178].

In the third place, the identification of different subsets of GSCs represents another controversial point. The GSCs are characterized by sustained self-renewal, persistent proliferation, tumor initiation, frequency within a GBM tumor, marker expression, ability to generate progeny of multiple lineages, and chemo/radio resistance [169,179,180]. There is no consensus for the surface markers used for identifying the GSCs, which are constantly updated [181]. The usual markers used for identification of GSCs tumoral subpopulations, are: CD133 (prominin-1), L1 cell adhesion molecule (L1CAM), CD44, CD90, A2B5, GPD1, CD49f, EGFR, CD184 [169,179,182,183,184]. Nuclear protein Ki-67, associated Nestin or *HOX* genes, MUSASHI-1 protein (translation regulator), KLF4, SALL4, OCT-4, GFAP [184] were also used for the GSCs characterization. The surface markers CD133, L1CAM, CD44, and the intracellular proteins and transcriptional factors, such as NANOG, NESTIN-neuroepithelial stem cell protein, Bmi1, SOX2, OLIG2, and MYC, overrun with those determined for recognition of NCS, thus challenging the recognition of GSCs in a mingled tumor [169,179,185]. Surface proteins (such as CD9, CD15, integrin-α6, enzymatic activity of aldehyde dehydrogenase 1 (ALDH1)) and signaling pathways (such as NOTCH, SHH, WNT/β-catenin, EGFR) are also overexpressed and overlap with those in NCS in terms of maintaining an undifferentiated character, a perpetual self-renewal state and a strong potency in initiating tumor development and proliferation [182,186,187]. All these markers reflect the GSCs’ high heterogeneity and try to define the GSCs’ multiple cellular subclones or multipotent microstates, which are inducing strong adaptability and high invasiveness in GBM [188]. Furthermore, GSCs have high plasticity and trans-differentiation capacity and rapidly transform into more aggressive phenotypes at the origin of intra-tumoral heterogeneity and therapy resistance [189].

In the fourth place, the continuous reshaping of GSCs hides their identity. The GSCs can also be characterized by their epigenetic, metabolic, and microenvironmental (or niche) parameters [190]. GSCs reside in four particular habitats inside GBM: perivascular, hypoxic, necrotic, and invasive [180]. The GSCs-TME bidirectional crosstalk, through tunneling nanotubes and tumor microtubes, allows multiple transfers of active molecules (e.g., mitochondria, calcium ions, oncogenic miRNA, RAS oncogenes, exosomes and other EVs) and reshapes the TME, remodeling also the immune system, the ECM/ stroma components, and the vascularization [189,191,192]. GSCs and endothelial cells reciprocally exchange miRNAs (e.g., miRNAs5096, miR-21, promoting angiogenesis) via EVs or GJs; miRNAs are also exchanged between ECM, GBM cells, and other cells (astrocytes, macrophages, and microglia) [193]. As the miRNAs modulate fundamental properties of GSCs, as self-renewal, proliferation, and growth [193], the miRNAs may represent a diagnostic tool for GBM and GSCs characterization [194].

Additionally, the complex and dynamic TME can rapidly modify or convert into another type of habitat in response to external stress conditions (e.g., chemo- or- immune- therapeutic intervention) [184]. Oppositely, the interaction with the TME also modifies the GSCs phenotype and even their stemness [182,190,192]. This constant dynamic induces the metabolic plasticity and survival of tumoral cells and directs the tumor growth, invasiveness, metastasis, recurrence, and resistance to chemo- or radiotherapy [181,195,196].

In the fifth place, the GSCs can transdifferentiate into tumor-supporting cells, as pericyte-like or endothelial cells, and can form functional tubular-like structures; this process is named vasculogenic mimicry. This trans differentiation mechanism is still unclear, but one of its inducing factors can be the chemotherapeutic stress (e.g., with temozolomide) [197]. It promotes tumor neovascularization and invasion, key factors of therapeutic resistance and tumor recurrence [170,197]. Therefore, we can sustain that GSCs are promoting themselves. The vasculogenic mimicry occurs mostly in aggressive glioma, primarily associated with hypoxic conditions, and can contribute to resistance to anti-angiogenic therapies [170,180,198].

Due to all these dynamics, no complete immunophenotype of GSCs has yet been identified with certainty [182]. Moreover, there are model limitations, too: we must consider, even from the beginning, that by performing an in vivo (serial transplantation) or an in vitro (cell culture) experiment, this will modify the original cell phenotype, distorting the GSCs identification through the use of cellular markers [155]. There are advantages and limitations of all GBM and GSCs models (cellular sorting by using surface markers, neurosphere culture, two-dimensional adherent culture, three-dimensional organoid culture on biomaterial scaffolds, genetically engineered mouse modeling, or patient-derived xenografts) [170]. All models studying the interactions between GSCs—GBM—immune system are unable to study the complete effects of an in vivo human immune system [170].

Finally, the further comprehension of the complex mechanics taking place between GSCS and TME could have a great beneficial influence on the therapeutic results of patients with GBM and may ensure the generation of new treatment strategies.

## 5. Tumor Microenvironment and Tumor-Associated Macrophages

The brain’s immunological system has several physiological and beneficial roles: phagocytosis of foreign substances, removal of cellular debris, tissue repair, axonal regeneration, synapsis plasticity [199].

The implication of the immune system in GBM’s tumor-associated macrophages (TAMs) is controversial, because, while the normal function of the immune system is to destroy the tumoral proliferation (several studies have already shown that microglia impair glioma invasiveness and growth in in vivo tumor models and organotypic slice cultures), alterations of microglia and macrophages (e.g., cytoskeletal reorganization) determine an opposite effect [200,201,202].

TAMs, along with their precursors, represent an essential part of the TME, accounting for the principal amount of the myeloid-line inflammatory infiltrate. TAMs form up to 30% of the tumoral mass, significantly exceeding the range of intratumor lymphocytes [203]. The number of TAMs has been reported to vary depending on the status of *IDH* mutation, noting that, when it comes to *IDH*-wild type GBM, the proportion is higher than in *IDH*-mutant GBM [204]. Considering the significant proportion of macrophages in the tumor mass, the understanding of the mechanisms involved in their activation, their interaction with tumor cells, and other elements in the TME are the basis for shedding light on novel prognostic factors and therapeutic targets.

TAMs arise from two different sources: microglia-brain tissue-resident—originating from the yolk sac, and macrophages—recruited from the circulation (as monocytes), in response to the release of chemoattractant molecules by tumor cells [205].

While in physiological conditions, only microglia are identified in the brain parenchyma, and in pathological conditions, the recruitment of monocytes from the periphery is prominent [206]. By studying the phenotype of CD11b+ cells (a marker of myeloid lineage), TAMs’ pleomorphism with a mixture of cells with pro- and anti-tumor effects was observed. It has been noted that monocytes initially have an anti-tumor effect, which then turns to a pro-tumor activity [207]. TAMs are distributed both intra- and peritumorally following the release of several chemoattractant factors by GBM cells (e.g., monocyte chemoattractant protein 1 (MCP1), granulocyte-macrophage colony-stimulating factor, colony-stimulating factor-1, and osteopontin) [208]. So far, it is not clear whether there are specific chemoattractant factors for monocytes and macrophages, respectively [209]. The expression of CD47 in tumoral cells inhibits the phagocytic function of TAMs, thereby providing a valuable pharmacologic target [206]. Hutter G et al. reported that the blockade of CD47 resulted in tumoral cells phagocytosis mediated by microglia, an effect independent from the presence of macrophages [210]. However, some evidence favors the maintenance of the phagocytic function of TAMs. Saavedra-López E et al. [211] revealed that the pseudo palisades found in GBM are composed of TAMs, which preserve their phagocytic activity and constitute a barrier towards GBM dissemination. Experimental studies also suggest that the anti-tumoral effect of TAMs in *IDH*-mutant GBM is mediated by ICAM-1/CD54 downregulation [212].

Although microglia are challenging to differentiate from macrophages (from a practical point of view), due to the expression of many common markers (e.g., IBA1, CD11b, CD45, CD68, CX3CR1), newly emerging data indicate the distinct role of the two types of cells in the TME [213]. Single-cell ribonucleic acid (RNA) sequencing revealed that the overall pro-inflammatory state of *IDH*-mutant GBM is associated with microglial function, while the anti-inflammatory environment of *IDH*-wild type GBM stems from macrophage activation [204]. The type of TAMs seems to vary with the tumor grade. In low-grade astrocytoma, the expression pattern of TAMs is mostly microglial-specific, while the higher-grade gliomas are correlated with the expression of macrophage-specific genes, possibly in conjunction with angiogenesis or modifications of the BBB [214]. Similar findings were reported by Friebel E et al. [215], who observed a progressive increase in bone marrow-derived macrophages in *IDH*-mutant GBM, *IDH*-wild type GBM and brain metastasis.

The localization of microglia and macrophages within the tumor is also different. Microglia accounts for the major population at the periphery of the tumor, as opposed to macrophages, which preferentially accumulate within the center [216]. Similar findings, obtained through single-cell RNA sequencing combined with multi-sector biopsies, were reported by Yu K et al., highlighting that microglia are located at the invasion front [217].

According to their phenotype and function, macrophages have been divided into classically activated M1, and alternatively activated M2, following in vitro exposure to different cytokines or microbial products. While the observed effects of M1 macrophages regard the pro-inflammatory and anti-tumorigenic roles, through interleukin (IL)-6, IL-12, IL-23, TNF-α synthesis, M2 macrophages ensure anti-inflammatory and pro-tumorigenic activity. The acquisition of the M1 or M2 phenotype depends on the cytokines expressed into the TME.

Therefore, the M1 phenotype is acquired by exposure to interferon-γ, lipopolysaccharide, or cytokines (TNF-α, granulocyte-macrophage colony-stimulating factor), while M2 polarization is acquired by IL-4, IL-10 and IL-13 exposure [218]. M2 macrophages are further divided in M2a, M2b and M2c, in concordance to the nature of the trigger and the markers expressed [206]. These two polarization states cooperate dynamically, the conversion from one to another being possible and even varying with the tumor progression (Figure 6). The enhancer of zeste homolog 2 oncogene is overexpressed in GBM, and its inhibition by specific microRNA was shown to switch in vitro the polarization of M2 macrophages to M1 [219]. The anti-CD47 strategy proved a similar effect [220].

Regarding their distribution within the tumor, Lisi et al. highlighted the presence of microglia/macrophages both intratumorally and at the periphery of GBM, by IBA1 immunostaining, with a negative correlation between expression intensity and median survival, suggesting its role as a potential prognostic tool. Intratumorally, M2 microglia/macrophages were significantly more numerous, as expressed by CD163 immunopositivity [221]. Tumor oxygenation is another factor on which the distribution of macrophages depends in hypoxic areas, M2 macrophages predominate, as compared to the oxygenated zones, where M0 and M1 are the most frequently encountered phenotypes [222]. However, there is an increased heterogeneity from one tumor to another regarding the proportion of anti-inflammatory and pro-inflammatory TAMs, respectively, which draws attention to the need for individual analysis of each case, and application of personalized medicine [204]. Furthermore, although the differentiation of macrophages into M1 and M2 polarization states is approachable in experimental studies, the translation of the methods into clinical research is not fully applicable, due to the diversity of these cells, and the multiple roles they play. These phenotypes may even coexist, and are not mutually exclusive [223]. However, recent studies pointed out debatable results related to the differentiation and polarization of macrophages [15,224].

There is also growing evidence supporting the existence of mixed types of TAMs, given the association of the pro-inflammatory environment with the immunosuppressive one in GBM [213]. Therefore, more investigation techniques (e.g., fluorescent-tagged monocytes in GBM animal models or thorough and extensive IHC analyses of macrophage markers in both polarized states), as well as additional models of great precision that reimagine the TME, are needed for an accurate perspective concerning this concept [15].

Exosomes are involved in intercellular communication, noting that those derived from GBM can induce the polarization of M1 or naive macrophages to M2. Subsequent exposure of GBM cells to exosomes derived from the reprogrammed macrophages has shown an increase up to 1000% in the migration capacity of tumor cells, contributing to the progression of the disease [225]. The growth and invasiveness of GBM are induced by TAMs, through different molecules, including stress-inducible protein 1, EGF, TGF-β, IL-6 [213,226]. The locomotion of the tumoral cells is enhanced by microglia through the PDGF receptor [227].

Another mechanism contributing to the progression of GBM ensured by TAMs is their involvement in angiogenesis. One mechanism is IL-6 secretion, with further JAK-STAT activation and endothelial cells’ progenitors’ recruitment. Perivascular TAMs were shown to be positively correlated with the capillary density in GBM and VEGF-A [228]. Through a 3D in vitro model, which aimed to replicate the in vivo immunosuppression and angiogenesis condition in GBM, it was observed that M2 macrophages promoted the endothelial cells’ proliferation and novel vessel formation by the interaction with integrin (αvβ3) receptors and Src-PI3K-YAP signaling [229].

As in other malignancies, cancer stem cells are known to promote tumor initiation, invasion, and local relapse. In addition to the communication with the tumor cells, TAMs and cancer stem cells interact, induce the secretion of TGF-β1 and favor the conversion of M1 to M2 phenotype [230]. Periostin is a chemoattractant for TAMs, preferentially synthesized by GSCs, whose level in GBM samples correlates with TAMs’ density. At the same time, it favors the harvesting of M2 macrophages, and by periostin silencing, the inhibition of tumor growth, and the improvement of in vivo survival are achieved [231]. One of the links between cancer stem cells and TAMs is the synthesis of C-C motif chemokine ligand 8 by macrophages, which has favored tumor growth and the aggressiveness of GBM in vivo [232].

To summarize, TAMs mediate the communication between tumor cells and other elements of the TME, sustaining key steps in GBM development. Ongoing research in this emerging field may ensure new advances in risk stratification for patients with GBM and may allow the development of novel therapies.

## 6. Other Immune Cells in Tumor Microenvironment and New Techniques for Cellular Mapping

As was illustrated above, the GBM tumor cells are highly heterogeneous and each cellular profile is characterized by its specific genotype and phenotype expression. However, the bidirectional interactions with TME are inducing a dynamic profile that masks critical differences and influences tumor onset, progression and therapeutical responses [214].

The TME is highly immunosuppressive [215] and the dynamic crosstalk immune cells—cancer cells—TME alters the activation of immune rejection mechanisms [232]. In order to identify the specific targets for efficient immunotherapy, it is fundamental to understand this dynamic crosstalk [233].

For characterizing the cellular heterogeneity, identifying novel cellular subsets and mapping the landscape of immune cells in GBM, the single cell-based techniques represent the basic tools. Data for a more accurate analysis and understanding of both the TME and immunology are brought by techniques based on single-cell RNA sequencing [232], as are the single-cell Tumor–Host Interaction tool (scTHI) [233], and high-dimensional single-cell profiling (Cy-TOF) (cytometry by time-of-flight, CyTOF) [215].

Next to the microglia and macrophages, other immune cells encountered in GBM’s TME are the MDSCs—the polymorphonuclear type (PMN-MDSCs, similar to neutrophils) and the monocytic type (M-MDSCs, similar to monocytes), the lymphocytes (T cells and occasional B cells), the NK cells, and the DC [234,235].

### 6.1. Aspects of Neutrophils’ Involvement in GBM

Neutrophils (PMNs) exert different destruction mechanisms (phagocytosis, cytotoxic granules releasing, ROS and nitrogen species, and extracellular traps) [236]. Even if neutrophils have a central role in the inflammatory process, they are also involved at different levels in the oncogenic process (tumor initiation, proliferation and dissemination). Neutrophils facilitate the tumor initiation through multiple mechanisms: they produce oxidative stress, can induce angiogenesis, attenuate the immune system (by the inhibition of macrophages’, DC’ and NK cells’ function), produce MMP9 and facilitate the extravasation of tumor cells [237,238,239].

The tumor-associated neutrophils (TAN) express two interchanging phenotypes during tumor progression: the anti-tumoral N1 and the tumor-promoting N2 [236,238]. The highly dynamic TME remodels the bone marrow myelopoiesis and induces the tumor-supportive phenotype [240].

Ferroptosis represents another neutrophilic mechanism, able to favor the oncogenic process. Neutrophils can transfer myeloperoxidase-containing granules to tumor cells, increasing ROS species levels, and inducing an iron-dependent accumulation of lipid peroxides; this will cause the necrosis of the tumor cells, with the attraction of more neutrophils [241], finally resulting a pro-tumorigenic positive feedback loop, amplifying necrosis development in GBM, and associated with poor survival [242].

The number and activity of tumoral and peripheral blood PMNs are correlated with GBM grade and with the survival prognosis [235,236]. In GBM there is also a high neutrophils-to-lymphocytes ratio (NLR) that characterizes the peripheral blood; NLR correlates with increased glioma grade, and it is associated with poor overall survival, and therapy resistance [243,244,245].

Neutrophils and PMN-MDSCs are different at biochemical, genomic, and functional levels [235]. In GBM patients, the majority of the MDSCs are of PMN-MDSCs type, expressing a phenotype of immature neutrophil. In human GBM, it is still unclear which specific subset of MDSCs is predominating [235,243]. Even the complete characterization of TAN plasticity (N1/N2 ratio) in GBM is still lacking [243,244]. The treatments targeting MDSCs or combined neutrophil-targeting therapy with other anticancer therapies have shown an increased survival rate in GBM patients [243].

### 6.2. Aspects of Dendritic Cells’ Involvement in GBM

Dendritic cells are ubiquitarian antigen-presenting cells, which initiate and maintain immune responses in peripheral lymphoid organs. DCs engage in an antigen-specific T cell differentiation by inducing deletion, anergy or regulation of regulatory T cells (Treg) [246], followed by activation, proliferation, and differentiation to effector cells: the cytotoxic T- and helper T lymphocytes [247,248].

There is also a polarized phenotype of DCs (e.g., for neutrophils), termed conventional or classical DCs (cDCs). The two cDCs, defined based on ontogenetic and phenotypic criteria, are cDC1s (induces TH1 stimulation) and cDC2s (induces TH2 responses) [247,249].

In normal conditions, DCs are present in the meninges, choroid plexus, and perivascular space [250], but not inside cerebral parenchyma [235]. In GBM patients, glioma-infiltrated DCs are reduced in number, even in the peripheral blood [236,251]. The cDC1s are recruited to the TME and can exert various anti-tumor mechanisms [252]; they may boost the anti-tumor activity of the T and NK cells [253].

The TME of GBM affects DCs through multiple pathways. DCs derived from tumors induce regulatory T-cells (Tregs) and suppress proliferation of cytotoxic T lymphocytes and NK cells. The suppression of DC maturation, and the consequent decrease in effector T cells activation facilitate the immune escape of glioma cells [247,254].

DCs’ characterization in CNS was not yet completed, and many DCs’ functions in GBM’s TME are still unsolved [255].

One direction in DC-based immunotherapies is the attempt to amplify cDC1′s tumor recognition [255]. Another is based on immunological memory: a DC vaccine (DCV) can initiate an anti-tumoral T-cell response, and a selective killing of the tumor cells [249,256]. If theoretically, a personalized DCV can prevent a tumor recurrence [256], the clinical response to DCV immunotherapy in GBM patients is variable (from no response to significant response) [248]. However, there are promising effects of DCV, such as the Tregs reduction in relapsed patients with high-grade glioma [251].

### 6.3. Aspects of Lymphoid Cells’ Involvement in GBM—T Lymphocytes and NK Cells

T lymphocytes are representing the majority of the tumor-infiltrating lymphocytes (TILs): cytotoxic T lymphocytes (CTLs), CD4^+^ T helper cells, and Tregs (CD4^+^/FoxP3^+^) [255].

Through several mechanisms (tolerance, anergy, senescence, and exhaustion) the GBM’s TME induces a global state of T-cell impairment: exhaustion of T cells, reduced effector functions, and increased surface expression of co-inhibitory immune checkpoints [255,257]. Human GBM’s TILs express the regulatory pathways of multiple immune checkpoints [258]; the checkpoint utilization is one of the several immunosuppression mechanisms shared by MDSCs and Tregs [248]. A novel mechanism of T cells inhibition in GBM is represented by T cell sequestration in the bone marrow [259].

The immunosuppressive TME blocks the cytotoxic response of CD8 T lymphocytes and increases the T cell tolerance by the expansion of Tregs [251,260]. GBM produces factors that actively recruit Tregs [255]. In high grade gliomas compared to lower grade, the number of cytotoxic TILs is generally reduced, and the Tregs number increases; the combination of those two factors tends to predict GBM patient survival [261].

NK cells are innate lymphoid cells and exert immuno-modulatory functions through the production of cytokines and the crosstalk with monocyte/macrophages, DCs, B and T lymphocytes, during the general immune response to GBM tumor [234].

NK cells are also displaying cytotoxic activities against “non-self” target cells, pathogens, and tumors, through several mechanisms: expression of perforin and granzymes, interactions of cell death receptor and/or antibody-dependent cellular cytotoxicity [262,263].

NK cells represent the least numerous populations in the GBM’s TME (about 2% of immune infiltrating cells, CD3^−^, CD56^+^), varying by the glioma subtype [264]. In glioma patients compared to healthy controls, the blood circulating NK are reduced in number [265].

Although NK cells act directly as antitumor agents, the clinical studies showed controversial results. This can be caused by different detection methods of tumor NK cells, leading to conflicting results [264]. In GBM patients, the lack of NK cells was associated with an increased probability of oxidative stress [236]. Furthermore, in a gastric cancer study, the intratumorally NK cells (CD57^+^) were associated with poor outcomes [266]. Another study showed that the pharmacologic impairment of autophagy functions acts as an immuno-modulator and promotes the genetically-engineered human NK cells into tumor sites, resulting in effective anti-GBM activity [267]. Therefore, more studies are required in order to determine the intra-tumoral impact of infiltrated NK cells and their therapeutic efficacy.

γδ T cells are a specialized subtype of T cells with one γ (gamma) and one δ (delta) chain made T-cell receptor (TCR). γδ T cells combine innate and adaptive (of their TCR and pleiotropic effector functions) immune properties and contribute to tumor immunosurveillance [268,269]. The γδ T cells–ligands interaction is not MHC-restricted, independent of antigen processing [270]. Similar to NK cells, γδ T cells directly destroy by recognizing tumor-associated antigens, and also facilitate the function of other immune cells (DCs, B cells and CD8^+^ T cells) [271].

γδ T cells are expressing more subpopulations in the blood and the tumor tissues [271]. GBM patients have an increased number of circulating Vδ1T cells (with immunosuppressive functions) and significantly decreased circulating Vδ2T cells (with cytotoxic activities) [236,272].

γδ T cells can be used for adoptive transfer in anti-tumor immunotherapy, and genetically engineered γδ T cells, in combination with checkpoint inhibitors, can have a better efficacy [271].

In total, the severe immunosuppressive TME and the lymphopenia are making the GBM tumors remarkably resistant to immunotherapy [273], and new treatment strategies can combine strategic timing of chemotherapy and immunotherapy (genetically modifying γδ T cells or NK cells) in order to achieve a significantly greater response.

Reaching higher dimensions in an immunological analysis is a huge challenge. Even if the bioinformatics is rapidly developing, the differential analyses for classification of cellular subsets are still based on the number of the features that can be measured (the cytometry techniques raised the detection limit over 50 parameters/cell) [274]. The existent data analysis capacity imposes the use of reduction/clustering dimensionality methods (e.g., the selection of lineage-specific markers). The application of several analytical frameworks also allows the detection of cellular ontogeny and developmental trajectories [274].

By analyzing the specific gene expression, ESTIMATE (Estimation of Stromal and Immune cells in Malignant Tumors using Expression data), a new algorithm for characterizing the TME profile, was used in order to evaluate the infiltration of stromal and immune cells and to stratify the GBM patients with distinct survival outcomes; a Prognostic Microenvironment-related Immune Signature (PROMISE model) for glioma was developed [275]. This model can be used for the development of new therapeutic targets and prognostic biomarkers.

We can sustain that, especially in the GBM, the heterogeneity of GAMs and myeloid cellular subsets and their functional expression still remain a puzzling target [207,236] and the immune cell landscape is still not completely mapped [234]. The improvement in our understandings of the complex immune crosstalk between GAMs—other myeloid cells (DCs and neutrophils)—lymphocytes (mostly cytotoxic T lymphocytes), can be applied for innovative therapies.

## 7. Cell-Targeting Therapies

Cellular heterogeneity, with different evolutionary forms and with different treatment behaviors, is responsible for the low survival rate of GB at 5 years (<10%) [276]. Understanding the cellularity and the basic molecular biology is essential for future therapies in GB.

To improve the GBM diagnosis and the therapeutic prediction value, it is crucial to develop methods more effective in isolating and identifying GSCs [184]. Cell sequencing and the single-cell RNA for sequencing the GSCs type with a distinct transcriptomic signature, the Molecular Imaging of TME, the TME High-Throughput Multiplex Immunohistochemical Imaging (mIHC) (based on brightfield IHC), the Nanostructured Probes, the Liquid Biopsy represent all newly emerged molecular diagnostic tools. These new detection techniques, the circulating tumor cells, the cell-free DNA, the circulating miRNAs, and the exosomes, all can have potent clinical applications as new biomarkers for non-invasive cancer diagnosis [170,193].

Multiple new treatment strategies targeting the GCSs have been suggested in GSM treatment, including immunotherapy, metabolic dependencies, posttranscriptional regulation, modulation of the TME, and epigenetic modulation [170]. Furthermore, the old anti-angiogenic therapies need to be reconsidered [181]. Because the GSCs subclones can have a different therapeutical susceptibility [170], the combined targeting of GSCs and TME for therapy might have the ability to reverse the treatment resistance of GBM [195].

Understanding the immune cells and the checkpoint modulators in GBM’s TME could be applied for boosting the host immune responses and will lead to novel immunotherapeutic strategies in the anti-GBM fight. A novel therapeutic strategy is to target the antigen presentation of TAMs to recruit and (re)-activate anti-tumoral effector T cells [277], or to increase TAM production of interferon and to recruit and activate T cells using a stimulator for the interferon gene (STING) agonist [278]. Slowing of glioma growth was obtained by using miR-142-3p, which influences TGF-β, or let-7b, which activates TLR7 in TAMs [279,280].

Another line for the new therapeutic approaches in GBM can focus on the factors driving the myeloid cells activity [237,238,239,242].

Immunotherapeutic approaches can be active and passive and can include anti-checkpoint inhibitors, TME remodeling, stimulation of tumor immunogenicity (DC vaccines, oncolytic viruses), and genetic engineering (adoptive T-cell therapies or chimeric antigen receptor T cell therapy); all those strategies aim to help the immune system to win the fight against cancer [248,249,255,256].

The tumor-associated astrocytes are studied in ongoing research, where the efficacy of IL 1β in attenuating GBM resistance in temozolomide therapy is determined. These astrocytes have a high expression of IL 1β compared with normal astrocytes, and RNAseq data identified some potential pathways for targeted therapies [281].

The computational data analysis capacity is still representing a limitation for the understanding of cellular differentiation and profile in the highly dynamic TME of GBM. Future research areas should direct toward combining both the machine learning methods and the investigator experience and requirements for obtaining a comprehensive understanding of TME, cells, and the immune system.

## 8. Conclusions

Glioblastomas’ cellular variety describes the wide range of molecular pathways involved in the tumor’ growth and progression. The fact that the GBM microenvironment modulates the biologic status of the tumor with the increase in its evasion capacity to treatment, implies the establishment of new personalized therapeutic strategies to fight cancer. A deeper understanding of cell interactions in TME and with the tumor could be the basis for these novel designed therapies.

## Figures and Tables

**Figure 1 cancers-14-01092-f001:**
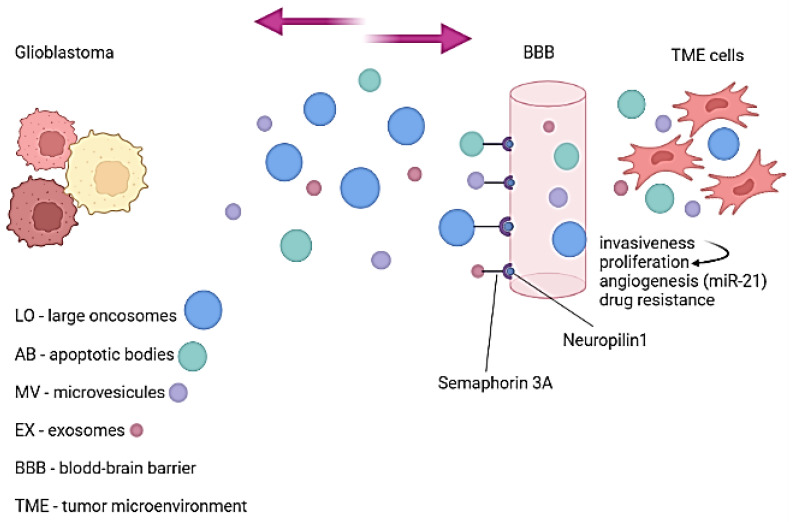
Intercellular communication between GBM and TME—direct exchange via EVs. EVs transport different cargo between GBM cells and TME, passing through the BBB by disrupting it. This is due to Semaphorin3A present on their surface that binds to its receptor on the BBB, the neuropilin1.

**Figure 2 cancers-14-01092-f002:**
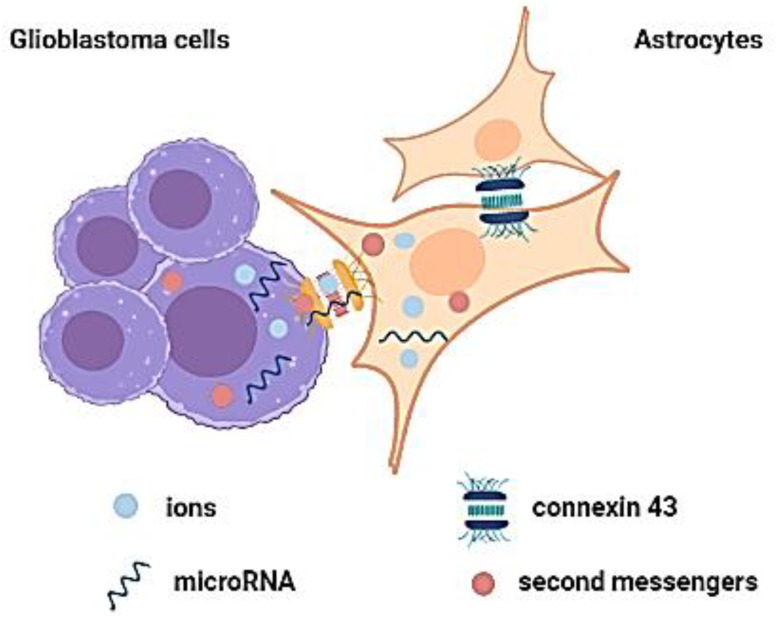
Intercellular communication between GBM and TME—direct exchange via GJs between glioma cells and astrocytes. When CX-43 increases, it leads to GBM cell proliferation and migration, overall enhancing chemotherapy resistance.

**Figure 3 cancers-14-01092-f003:**
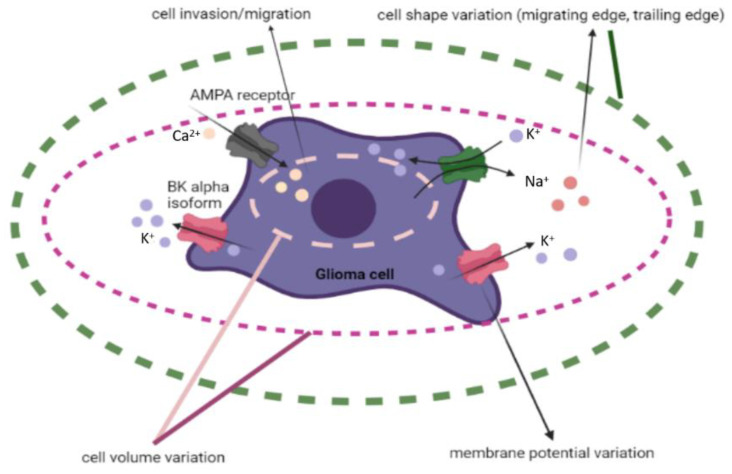
Intercellular communication between GBM and TME—direct exchange via ion channels (it is leading to cell volume and shape variations or loss of epithelial polarization by ion concentration changes).

**Figure 4 cancers-14-01092-f004:**
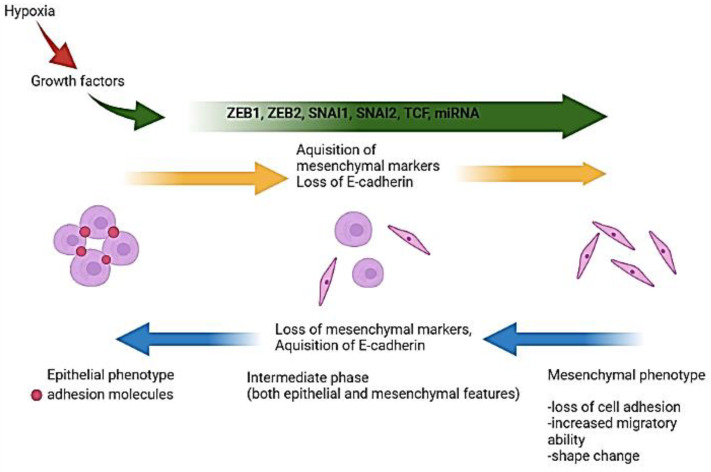
Epithelial to mesenchymal transition of GBM cells passing through an intermediate phase where both epithelial and mesenchymal features are present. Note that this process is reversible, and it implies acquisition and loss of markers.

**Figure 5 cancers-14-01092-f005:**
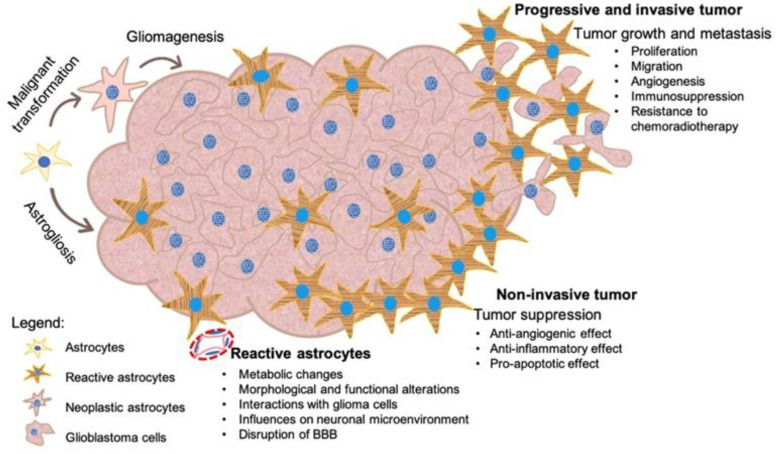
Pathogenic mechanisms underlying the role of reactive astrocytes in GB development and progression. Astrocytes can undergo malignant transformation to promote gliomagenesis, but can also become activated, as components with essential influence on the TME. Tumor-associated reactive astrocytes exhibit morphological, functional, and metabolic changes and, by their interaction with glioma cells, can either suppress or promote tumor maintenance and invasion, as well as resistance to chemo-radiotherapy.

**Figure 6 cancers-14-01092-f006:**
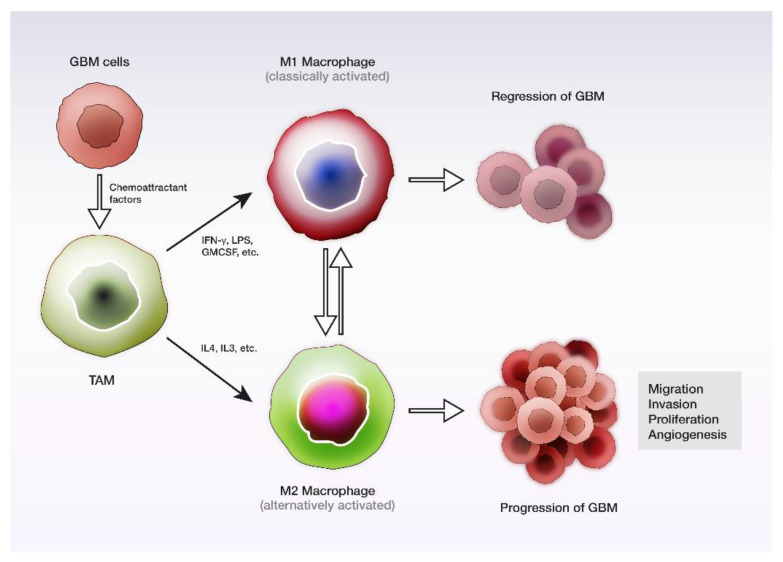
GBM cells secrete chemoattractant factors through which they recruit TAM. By the two polarization states, M1 and M2, TAM may trigger regression or progression of GBM.

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
