# Peer review of "Glioblastoma Microenvironment and Cellular Interactions"

_cancers, 2022, doi:10.3390/cancers14041092_

Round 1
Reviewer 1 Report
In this manuscript, the authors review the current understanding on glioblastoma microenvironment, and summarize the interactions between tumoral/nor-tumoral cells and tumor microenvironment. Overall, it is written well, and would attract the attention from researchers who are interested in or working on studying glioblastoma tumor microenvironment.
Several minor issues need to be fixed.
- In line 73, the size of large oncosomes should be > 1μm.
- In Figure 1, “microvesicules” should be “microvesicles”; and “blodd-brain barrier” should be “blood-brain barrier”.
- In line 384, “Gln is produced by tumor cells” sounds like the only source of Gln is from tumor cells. The authors may consider altering it.
- Higher resolution of pictures are required for figures.
Reviewer 2 Report
Cancers.
-
There are several issues with the references in the Introduction. Please check all references, especially in this part.
-
Line 34: reference 4-6 are not that related to this paragraph.
-
Line 33: WHO has updated their classifications in 2021. The authors should consider updating with new information and citing related papers.
-
Line 38: reference 8 looks like not related. Moreover, this reference it's too old and hard to find.
-
Line 59 and 67: reference 10 focused on miRNA, Not a good reference for this part.
-
-
The section "2. Intercellular communication between GBM cells and TME" is very long and included several essential parts. The authors should consider adjusting the structure and format of this part and include some sub-titles to provide a clear view.
-
Line 224-251: The authors reviewed the inconsistent expression of E-cadherin, but did not clearly discuss the cause of heterogeneous, including inter- and intra- tumor heterogeneity. TCGA molecular classifications already indicate patients could show different gene expression profiles (PMID: 20129251). Single-cell studies also showed that even inside the tumor, there is significant heterogeneity showing different proportions of cells characterized by different sub-class (PMID: 24925914). Please include these information and cite these papers.
-
The section "4. GSCs". The authors should be cautious because the concept of glioma stem cells is still controversial. The author spent more than two pages discussing it but didn't emphasize the controversy enough. The two major references (157-158) about the GSCs in this part both mentioned the controversies about the GSCs. I recommend the authors reduce the size of this part, especially about the GSCs communications and crosstalk between GSCs and TME. References in this part are not solid enough to solve the controversies. It could present misleading information for readers. Most importantly, the author should spend a paragraph discuss the controversies of GSCs.
-
While reducing the size of section 4, I think the section "5. Tumor microenvironment and tumor-associated macrophages" is not reviewed enough.
-
The microglia and macrophages are not reviewed clear enough. There are a lot of good reviews about microglia/macrophages. Like PMID: 26713745, PMID: 31697921, PMID: 20371344 to learn from.
-
The authors only included one reference about the single-cell study of TAMs (ref 189). The sentence in lines 586-588 should be cautious because it's from one study with a limited sample size and relatively small cell number. Single-cell studies are important to the field of TAM research, the authors should improve this part. There are a lot of very nice single-cell studies that provide more information related to this part and some previous sections. Including (PMID: 28360267, PMID: 34692159, PMID: 32470397, PMID: 33608526, etc.). These papers also provide additional information on the topics in this manuscript. For example, about the TAM distribution within the tumor in line 606, PMID: 34692159 presented the cell distribution with single-cell RNA-seq from multi-sector biopsies. Also, other single-cell studies focused on the cross-talk between glioma cells and TME cells (PMID: 33155039), which is also very important for this manuscript. The author should discuss the information from these studies and cite these papers.
-
-
For Figures 1-4, the descriptions are not clear. Please provide more detailed descriptions in figure legends.
-
The abstract is relatively short and does not highlight all the key information in this manuscript. The authors should consider extending or rewriting the abstract.
Minor issues:
-
Line 40: TEM should be TME.
-
Line 70: EVs, should use the full term just like the beginning of line 101, because this is just like a sub-title of this big paragraph. This suggestion is also applicable to "GJs" in line 107.
-
Line 71: What is "genomic material" here?
-
Line 73: Please correct ">1m"
-
Line 102: What is "Gjnal". Lines 107-130, please double-check the usage of the term "GJs", someplace were using "GJs" and someplace were using "Gjs".
-
Figure 5, label "Tumor progression and invasion", it could be better to use the noun-form term just like the other label "Non-invasive tumor"
-
Gene names and protein names in the manuscript should use the right nomenclature. For example, gene names in Line 369, p53 should be "TP53", Line 587-588, "IDH" should be "IDH
Reviewer 3 Report
The manuscript entitled „Glioblastoma microenvironment and cellular interactions” by Carmen-Bianca Crivii et al. summarises selected aspects of cellular communication in regards to glioblastoma multiforme development. The article is interesting, extensive, and valuable. Nevertheless, in my opinion, it needs some corrections to make the content easier to understand. Some of the suggestions are listed below:
Major issues:
- Paragraph 2 – Intercellular communication between GBM cells and TME:
- Figure 1 – the text of the manuscript points to an involvement of VEGF in the effects exerted by EVs. However, it is not illustrated in the given figure. Please correct.
- Lines 95-98: it is not clear if hypoxia is involved in EVs driven impact on the PD1/PD-L1 axis. Please specify.
- Gap junctions: lines 118-120. Authors point to an involvement of connexin-43 in GBM development and resistance to therapy, however, in a further sentence they point to downregulation of this protein in higher grade glioma. That seems to be opposed to the expected effect. Please explain.
- A part summarizing EMT. The previous part of the paragraph focuses on cellular elements, which condition cell-cell communication. Including the EMT process described in this part of the article, causes the manuscript to lose its logical flow and it is difficult to understand. Please rebuild this part.
- Paragraph 3 – Tumoral and reactive astrocytes:
- Lines 269-273 contain more general information than lines placed above (259-268). That makes an introduction to more detailed descriptions a bit confusing. Please consider placing information in the reverse order.
- The heading of the paragraph suggests occurring differences between tumoral and reactive astrocytes. However, the content of this paragraph does not, in any way, differentiate between the two subtypes of astrocytes. Please point out the differences between them.
- Lines 298-299 – they suggest pointing to any mechanisms exerted by immune cells and cancer cells, that activate astrocytes. But further part (lines 299-311) focuses on morphological changes in astrocytes and says nothing about mechanisms via which they become activated. Please, consider replacing information and ensure specific shreds of evidence on the role of cancer/immune cells in astrocytes activation.
- Lines 343-354: authors want to focus on cell-cell communication and interaction, but in this part, they describe rather a role of astrocytes in disease-associated symptoms, which deviates from the main topic of the article. Please consider, either erasing this part, rebuilding it, or putting it at the end of the paragraph, to make the content more logical.
- Paragraph 4- glioma stem cells (GSCs):
- .
- Here again, the authors digress from the main subject of communication between tumor cells and TME. It seems that the main data starts in line 498. Please, consider shortening or rebuilding the previous information, to stay closer to the main topic of an article, and thus facilitate the reception of the content.
- General comments:
- .
- In the introduction, the authors point to communication between glioma cells and the number of immune cells (macrophages, natural killer, cytotoxic lymphocytes, dendritic cells). However, the further part of the article focuses only on communication between macrophages and tumor cells (paragraph 5), excluding other immune cells. An addition of another section, which at least in short will summarize the role of remaining immune elements in GBM development, and illustrate the issue in detail, should be considered.
- In a few parts authors suggest, that described interactions have the potential to become new goals for antitumor therapy. Please consider enriching your manuscript with a short paragraph that will gather available data, supporting your thesis.
Minor issues:
- Introduction – line 40. There is a typo in an acronym standing for microenvironment. Please correct.
- The part about gap junctions (from line 101 and further) - the abbreviations that appear are not uniform - they appear once in capital letters, then in small letters. Please standardize.
Round 2
Reviewer 2 Report
The authors have addressed all my concerns and did proper modifications to the manuscript. The current version of this manuscript improved a lot. I recommend publication.
There is one minor issue that should be fixed in Figure 3. The "2+" or "+" are not using superscripts. Please correct.
Reviewer 3 Report
The authors referred to all comments, rebuilt, and enriched the article with suggested paragraphs. In my opinion, the article in this form can be accepted for publication.